# Does weighting improve matrix factorization for recommender systems?

## Abstract

Matrix factorization is a widely used approach for top-N recommendations and collaborative filtering. When it is implemented on implicit feedback data (such as clicks), a common heuristic is to upweight the observed interactions. This strategy has been shown to improve the performance of certain algorithms. In this paper, we conduct a systematic study of various weighting schemes and matrix factorization algorithms. Somewhat surprisingly, we find that the best performing methods, as measured by the standard (unweighted) ranking accuracy on publicly available datasets, are trained using unweighted data. This observation challenges the conventional wisdom in the literature. Nevertheless, we identify cases where weighting can be beneficial, particularly for models with lower capacity and certain regularization schemes. We also derive efficient algorithms for minimizing a number of weighted objectives which were previously unexplored due to the lack of efficient optimization techniques. Our work provides a comprehensive analysis of the interplay between weighting, regularization, and model capacity in matrix factorization for recommender systems.

## Keywords

Recommender Systems, Collaborative Filtering, Matrix Factorization

**ACM Reference Format:**

Anonymous Author(s). 2024. Does weighting improve matrix factorization for recommender systems?. In . ACM, New York, NY, USA, 9 pages. https://doi.org/10.1145/nnnnnnn.nnnnnnn

## 1 Introduction

Recommender systems are an integral part of the Web. In a typical recommender system, we observe how a set of users interact with a set of items, and our goal is to recommend each user previously unseen items that they would like. In this paper we consider user interactions (click, purchase, viewing, etc.) as implicit feedback (Hu et al. [14]). Unlike explicit feedback (ratings), implicit feedback is organically generated as part of the users' interaction with the recommender systems, and hence more easily accessible and prevalent.

Traditionally, matrix factorization has been the go-to choice for modeling implicit feedback data (Hu et al. [14], Liang et al. [17], Pan et al. [21], Rendle et al. [24], Steck [27]). At its core, matrix factorization involves decomposing the user-item interaction matrix into a product of two matrices, capturing latent patterns among users and items via low-dimensional representations. The simplicity and robustness offered by matrix factorization techniques makes them a strong baseline in the literature. Even with the recent emergence of the powerful deep-learning-based approaches, simple linear models, including matrix factorization, when carefully tuned, can still stay competitive or even outperform their neural network counterparts (Ferrari Dacrema et al. [10], Rendle et al. [25], Steck [30]).

A unique challenge presented with implicit feedback for matrix factorization is how to handle the zeros in the user-item interaction matrix. Intuitively, we expect the interacted items to convey more information about a user's preference. The reason a user did not interact with a particular item could be either a lack of interest or, more likely, a lack of awareness/exposure (Liang et al. [17]). Consequently, a common approach used in the literature is to weight the data differently based on whether their value is zero or one ([14, 21]), specifically, up weighting the interactions (ones) to reflect our prior belief that we are more certain about these observations. Empirically it has been demonstrated that such weighting generally helps with recommendation accuracy in previous studies (Hu et al. [14], Liang et al. [17], Pan et al. [21], Steck [27]).

Curiously, a notable exception to this weighting strategy is the EASE model (Steck [30]), which learns an item-item similarity matrix from unweighted data. Even though EASE model is a full-rank model, as opposed to the low-rank matrix factorization which is the main focus of this paper, the fact that EASE can outperform not only the weighted matrix factorization, but also powerful deep-learning-based approaches suggests there is more to be uncovered here.

In this paper, we take a systematic look at various weighting schemes and matrix factorization algorithms and find that, somewhat surprisingly, the conventional wisdom on up weighting the observed interactions carries more nuance. Specifically, we observed that the best performing methods, as measured by (unweighted) ranking accuracy on publicly available datasets, are generally trained using unweighted data. On the other hand, there are cases where weighting can be beneficial, especially for models with lower capabilities. This is in parallel with observations made with deep models (Byrd and Lipton [5]).

We highlight the following contributions of our paper:

(1) We provide experimental evidence that unweighted methods outperform their weighted counterparts on standard benchmarks in recommender systems, which undermines the conventional wisdom in the literature.

(2) We derive efficient algorithms for minimizing weighted objectives for various matrix factorization models. To the best of our knowledge, most of these objectives were never studied in the

literature due to the lack of efficient optimization algorithms. Our work fills in this gap.

(3) We systematically study the interplay between weighting, regularization, and model capacity by running exhaustive experiments over a wide range of methods.

## 2 Preliminaries

In this paper we consider the problem of top-N recommendation [9, 19] with implicit feedback data [14, 21]. We assume access to a matrix (typically sparse and binary) of user-by-item interactions $X \in \mathbb{R}^{|\mathcal{U}| \times |\mathcal{I}|}$ where $\mathcal{U} = \{1, 2, \cdots, U\}$ and $\mathcal{I} = \{1, 2, \cdots, I\}$ denotes the set of users and items respectively. If $X_{ui}$ is positive, then we say a user $u$ has interacted with an item $i$. If $X_{ui}$ is zero, then we say a user $u$ has no interaction with item $i$.

Following the standard settings [10, 18, 30], we "binarize" the user-by-item interaction matrix so that all the positive entries of $X$ take value 1 and the zero entries of $X$ remain unchanged. This is done to facilitate the use of standard ranking metrics such as the truncated normalized discounted cumulative gain (NDCG@$R$) ([15]) and Recall@$R$. In top-N recommendation, we aim to provide $N$ recommendations for each user and our goal is to have the heldout items an user actually interacted with rank as high as possible.

We outline a few important model building blocks that will be used throughout this paper, starting with the standard weighted matrix factorization:

**Weighted matrix factorization.** In weighted matrix factorization (WMF) (Hu et al. [14], Pan et al. [21]), we learn $d$-dimensional user and item factors $U \in \mathbb{R}^{|\mathcal{U}| \times d}$, $V \in \mathbb{R}^{|\mathcal{I}| \times d}$ via the following objective:

$$\min_{U,V} \left\| \sqrt{W} \odot (X - UV^T) \right\|_F^2 + \lambda \|U\|_F^2 + \lambda \|V\|_F^2, \quad (1)$$

where $d \leq |\mathcal{I}|$, $\lambda$ is the regularizer strength, and $W \in \mathbb{R}_{>0}^{|\mathcal{U}| \times |\mathcal{I}|}$ is a set of positive weights. Typically $W$ is set such that the weights for $X_{ui} = 1$ is larger than the weights for $X_{ui} = 0$, for example $W_{ui} = (\alpha - 1)X_{ui} + 1$ for $a > 0$ as used in Hu et al. [14], Pan et al. [21], and Steck [27]. With such a weighting function, WMF can be efficiently optimized via alternating least squares (Hu et al. [14]). In this paper, we follow the same weighting function in all our experiments.

**Asymmetrical (weighted) matrix factorization.** Asymmetrical matrix factorization (AWMF) Paterek [22] is a closely-related approach with the following objective[1]:

$$\min_{U,V \in \mathbb{R}^{|\mathcal{I}| \times d}} \left\| \sqrt{W} \odot (X - XUV^\top) \right\|_F^2 + \lambda \|U\|_F^2 + \lambda \|V\|_F^2 \quad (2)$$

Note that with AWMF, both of the learned matrices $U$ and $V$ are of the size $|\mathcal{I}| \times d$. Comparing the first term in eq. (2) with eq. (1), we can interpret AWMF as having a constrained latent user factor. Specifically, the user factor for user $u$ in AWMF is implicitly defined as the sum of the latent item factors (in $U$) of the interacted items by user $u$. Assuming $|\mathcal{U}| \gg |\mathcal{I}|$ (which is often the case in many recommendation scenarios), compared to an unconstrained latent user factor that has to be learned for each individual user in the

---

[1]The original A(W)MF of Paterek [22] does not have weights.

WMF model, the constraint results in a considerable reduction in the number of model parameters. AWMF also has strong connections to autoencoder models (Liang et al. [18], Steck [28]).

Another variate of AWMF optimizes the following objective:

$$\min_{U,V \in \mathbb{R}^{|\mathcal{I}| \times d}} \left\| \sqrt{W} \odot (X - XUV^\top) \right\|_F^2 + \lambda \|UV^\top\|_F^2 \quad (3)$$

The unweighted version of eq. (2) and eq. (3) are studied in Steck [31]. It is noted that eq. (2) is a "weight-decay" style regularization, while eq. (3) corresponds asymptotically to a "dropout/denoising" style regularization [6]. The results in Steck [31] demonstrate that the unweighted version of eq. (3) tends to perform better with a fairly sizable margin.

Finally, we consider a full-rank version of eq. (3):

$$\min_{B \in \mathbb{R}^{|\mathcal{I}| \times |\mathcal{I}|}} \left\| \sqrt{W} \odot (X - XB) \right\|_F^2 + \lambda \|B\|_F^2. \quad (4)$$

This is similar to the objective of EASE Steck [30], except that we introduce a weighting matrix $W$ and drop the zero-diagonal constraint. Note that we made the latter choice as to be consistent with the other models for a fair comparison, even though this results in slightly reduced accuracy as shown in Steck [29]. We expect the full-rank model to generally outperform its low-rank counterparts.

Several further comments are in order:

- We choose the square loss (here $\| \cdot \|_F$ denotes the Frobenius norm), as is standard in the literature when applying matrix factorization to collaborative filtering/top-$N$ recommendations (Hu et al. [14], Ning and Karypis [19], Steck [30, 31]). De Pauw and Goethals [8], Liang et al. [18] observe that training with alternative loss functions such as the logistic loss yield better ranking accuracy over the *unweighted* square loss, as the logistic loss reweights the data (Ayoub et al. [2]). However, as our experiments demonstrate, training on unweighted data can give the best performance.

- We test various forms of $\ell_2$ regularization. As was observed by Steck [31], Steck and Garcia [32] and Jin et al. [16], for an item-similarity matrix with low rank, i.e. $B = UV^\top$, choosing the right regularization scheme can have a significant impact on ranking accuracy. Note that this adds an additional hyperparameter $\lambda$, which needs be selected — using a held-out validation set — separately for each regularizer.

Note that even though we explicitly write the weighting matrix in all of the objectives in this section, we can recover the unweighted versions by setting $W$ to be all 1's. For most of these objectives, specifically for eqs. (2) to (4) with general weighting matrix, the challenge lies in efficient optimization with large-scale recommendation datasets. To the best of our knowledge, none of these objectives have been studied in the literature. In the subsequent sections, we dive into the details on how we design optimization algorithms.

## 3 Optimization

In this section, we derive closed-form solutions for the optimization problems given in eqs. (2) to (4). In section 4, these solutions will be paired with the conjugate gradient method to produce efficient algorithms for the various optimization problems.

## 3.1 Closed-Form Solution: Unregularized

In this section, we show that the objective functions for learning the item similarity matrices $UV^\top$ and $B$ in eqs. (2) and (3) and eq. (4) can be solved in closed form when $\lambda = 0$. The choice to first present the results for the unregularized case is done solely to simplify the exposition, and the proceeding section contains the extension to $\lambda > 0$.

Our results refute a claim made by Jin et al. [16], Steck [29], specifically that a closed form solution to eqs. (2) to (4) does not exist for an arbitrary weight matrix $W \in \mathbb{R}_{>0}^{|\mathcal{U}| \times |\mathcal{I}|}$. Thus we resolve an open question by giving a closed form solution to the weighted objectives in eqs. (2) to (4). The derivation requires reviewing the definition of the Kronecker product.

*Definition 3.1 (Kronecker product).* The Kronecker product of matrices $A \in \mathbb{R}^{m \times n}$ and $B \in \mathbb{R}^{p \times q}$ is denoted by $A \otimes B$ and defined to be the block matrix

$$A \otimes B = \begin{bmatrix} a_{11}B & a_{12}B & \cdots & a_{1n}B \\ a_{21}B & a_{22}B & \cdots & a_{2n}B \\ \vdots & \vdots & \ddots & \vdots \\ a_{m1}B & a_{m2}B & \cdots & a_{mn}B \end{bmatrix}.$$

We refer the reader to Chapter 4.2 of Horn and Johnson [13] for a summary of elementary properties of the Kronecker product, some of which will be used below. We will also need the following "vectorization" lemma.

LEMMA 3.2 (LEMMA 4.3.1 OF HORN AND JOHNSON [13]). *Let matrices $A \in \mathbb{R}^{m \times n}, B \in \mathbb{R}^{p \times q}$ and $C \in \mathbb{R}^{m \times q}$ be given and $X \in \mathbb{R}^{n \times p}$ be unknown. The matrix equation*

$$AXB = C$$

*is equivalent to the system of $qm$ equations in $np$ unknowns given by*

$$(B^\top \otimes A) \operatorname{vec}(X) = \operatorname{vec}(C).$$

*That is, $\operatorname{vec}(AXB) = (B^\top \otimes A) \operatorname{vec}(X)$, where $\operatorname{vec}(A)$ is the $mn \times 1$ column vector obtained by stacking the columns of $A$ with the leftmost column on top[2].*

The vectorization lemma has historically been used to reason about and compute solutions to Sylvester's equation of the form $AX + XB = C$ and Lyapunov equation of the form $AXA^\top - X + B = 0$, which arise naturally when studying linear dynamical systems in control theory. In this context the vectorization lemma is used to "linearize" these seemingly nonlinear matrix equations, and our closed-form solution takes inspiration from this technique.

We will also employ the following proposition in our derivations, so as to avoid repetitive calculations and streamline the presentation.

PROPOSITION 3.3. *For matrices $A, W \in \mathbb{R}^{u \times i}$ and $B, C \in \mathbb{R}^{i \times d}$,*

$$\operatorname{vec}(A^T(W \odot (ABC^\top))C) = (C \otimes A)^\top \bar{W}(C \otimes A) \operatorname{vec}(B) \quad (5)$$

*where $\bar{W} = \operatorname{diagMat}(\operatorname{vec}(W))$. Furthermore, we also have that*

$$\operatorname{vec}((W^\top \odot (CB^\top A^\top))AB) = (AB \otimes I_i)^\top \hat{W}(AB \otimes I_i) \operatorname{vec}(C), \quad (6)$$

*where $\hat{W} = \operatorname{diagMat}(\operatorname{vec}(W^\top))$.*

---

[2]$\operatorname{vec}(A) = A$.flatten("F") in NumPy.

PROOF. We start the proof by showing eq. (5). Employing lemma 3.2, we have that

$$\operatorname{vec}(A^T(W \odot (ABC^\top))C)$$
$$= (C \otimes A)^\top \operatorname{vec}(W \odot (ABC^\top))$$
$$= (C \otimes A)^\top (\operatorname{vec}(W) \odot \operatorname{vec}(ABC^\top))$$
$$= (C \otimes A)^\top \operatorname{diagMat}(\operatorname{vec}(W)) \operatorname{vec}(ABC^\top)$$
$$= (C \otimes A)^\top \bar{W}(C \otimes A) \operatorname{vec}(B),$$

where the first and last equalities use lemma 3.2 and the third equality uses the fact that for vectors $x, y$, it holds that $x \odot y = \operatorname{diagMat}(x)y$.

Now we turn our attention to showing eq. (6). Again employing lemma 3.2, we have that

$$\operatorname{vec}((W^\top \odot (CB^\top A^\top))AB)$$
$$= (AB \otimes I_i)^\top \operatorname{vec}(W^\top \odot (C(AB)^\top))$$
$$= (AB \otimes I_i)^\top (\operatorname{vec}(W^\top) \odot \operatorname{vec}(C(AB)^\top))$$
$$= (AB \otimes I_i)^\top \operatorname{diagMat}(\operatorname{vec}(W^\top)) \operatorname{vec}(C(AB)^\top)$$
$$= (AB \otimes I_i)^\top \hat{W}(AB \otimes I_i) \operatorname{vec}(C),$$

where the first and last equalities use lemma 3.2 and the third equality uses the fact that for vectors $x, y$, it holds that $x \odot y = \operatorname{diagMat}(x)y$. This completes the proof. □

With these preliminary results in hand, we are ready to tackle eqs. (2) to (4) with $\lambda = 0$.

***Solving eq. (4), unregularized:*** We start by solving the optimization problem given in eq. (4) as it is the simplest to solve and thus provides the cleanest presentation of our analysis. We start taking its derivative and setting it to be zero,

$$X^T(W \odot (XB - X)) = 0.$$

Previous attempts at finding a closed form solution failed due to the presence of the Hadamard (element-wise) product $\odot$, which does not allow the order of operations to be exchanged. Employing operations to be exchanged. Employing eq. (5) of proposition 3.3 and rearranging, we can rewrite the above matrix equation as

$$\operatorname{vec}(X^\top(W \odot X))$$
$$= \operatorname{vec}\left(X^T(W \odot (XB))\right)$$
$$= (I_{|\mathcal{I}|} \otimes X^\top)\bar{W}(I_{|\mathcal{I}|} \otimes X) \operatorname{vec}(B) = H(B, 0) \operatorname{vec}(B),$$

where we defined $H(B, 0)$[3] $= (I_{|\mathcal{I}|} \otimes X^\top)\bar{W}(I_{|\mathcal{I}|} \otimes X)$. Rearranging the above equation gives the following closed form for $B$

$$\operatorname{vec}(B) = H(B, 0)^{-1} \operatorname{vec}(X^\top(W \odot X)). \quad (7)$$

We remark $H(B, 0)$ is guaranteed to be invertible so long as $X$ has full column rank, i.e. $X^\top X$ is invertible. Finally, we note the while a closed form solution for $B$ was given in terms of $W$ and $X$, this method can be used for finding a closed form solution for any optimization problem of the form

$$\min_{B \in \mathbb{R}^{|\mathcal{I}| \times |\mathcal{I}|}} \left\| \sqrt{W} \odot (X' - XB) \right\|_F^2, \quad (8)$$

---

[3]we slightly abuse notation here by writing H(B,0) since H is not a function of $B$ but is a function of 0.

which may be of independent interest. An interesting extension would be to leverage our analysis and derive a closed form solution for the case when $B$ is constrained to be a zero-diagonal matrix, as was done explicitly in Steck [30] and implicitly in Ning and Karypis [19]. We believe the use of the vectorization lemma will facilitate such an analysis and we highlight this as potential future work.

***Solving eqs. (2) and (3), unregularized:*** We now apply proposition 3.3 to eqs. (2) and (3) with $\lambda = 0$ using a similar procedure. Since we will employ alternating minimization for solving eqs. (2) and (3), we first fix $V$ and compute the closed form solution for $U$ and then do the converse. Taking the derivative of the expression to the left of the sum in eq. (3) with respect to $U$ and setting it to be zero gives

$$X^\top(W \odot (XUV^\top - X))V = 0 \, .$$

Employing eq. (5) of proposition 3.3 and rearranging the above display gives

$$\begin{aligned}
\text{vec}&(X^\top(W \odot X)V) \\
&= \text{vec}(X^\top(W \odot (XUV^\top))V) \\
&= (V \otimes X)^\top \bar{W}(V \otimes X)\,\text{vec}(U) \\
&= H_D(U, 0)\,\text{vec}(U) = H_W(U, 0)\,\text{vec}(U) \, ,
\end{aligned}$$

where $H_D(U, 0) = H_W(U, 0) = (V \otimes X)^\top \bar{W}(V \otimes X)$. We let $H_D(U, 0)$ denote the Gram matrix of unregularized AWMF with dropout with respect to $U$ and $H_W(U, 0)$ denote the Gram matrix of unregularized AWMF with weight decay. In later sections when we consider $\lambda > 0$, the utility of these shorthands will become more apparent. Rearranging the above equation gives the following closed-form expression for computing $U$

$$\text{vec}(U) = \left((V \otimes X)^\top \bar{W}(V \otimes X)\right)^{-1} \text{vec}(X^\top(W \odot X)V) \, . \quad (9)$$

So long as both $V$ and $X$ have full column rank, the inverse above is guaranteed to exist. Repeating the same calculations for $V$ and using eq. (6) of proposition 3.3, we get that

$$\text{vec}(V) = \left((XU \otimes I_{|\mathcal{I}|})^\top \hat{W}(XU \otimes I_{|\mathcal{I}|})\right)^{-1} \text{vec}((W^\top \odot X^\top)XU) \, . \quad (10)$$

Here $XU$ must have full column rank for the inverse to exist. In the future we will also use the shorthand $H_D(V, 0) = H_W(V, 0) = (XU \otimes I_{|\mathcal{I}|})^\top \hat{W}(XU \otimes I_{|\mathcal{I}|})$. This concludes our analysis of the case $\lambda = 0$, and we are ready for the generalization to regularization.

## 3.2 Closed-Form Solution: Regularized

In this section, we show that the regularized objective functions for learning $B$, $U$ and $V$ in eqs. (2) to (4) can be solved in closed form.

***Solving eq. (4), regularized:*** Taking the derivative of the optimization problem in eq. (4) and setting it to be zero, we get

$$X^T (W \odot (XB - X)) + \lambda B = 0 \, .$$

Rearranging the above equation and employing proposition 3.3, we write

$$\begin{aligned}
\text{vec}&(X^\top(W \odot X)) \\
&= \text{vec}\left(X^T (W \odot (XB)) + \lambda B\right) \\
&= H(B, 0)\,\text{vec}(B) + \lambda(I_{|\mathcal{I}|} \otimes I_{|\mathcal{I}|})\,\text{vec}(B) \\
&= H(B, \lambda)\,\text{vec}(B) \, .
\end{aligned}$$

Thus we get the following closed form expression for $B$ when solving eq. (4) for $\lambda > 0$,

$$B = H(B, \lambda)^{-1}\,\text{vec}(X^\top(W \odot X)) \, . \quad (11)$$

***Solving eqs. (2) and (3), regularized:*** We now optimization problems in eqs. (2) and (3), we first fix $V$ and find the closed form expressions for $U$. Taking the derivative of eq. (3) with respect to $U$ and setting it to be zero gives

$$X^\top(W \odot (XUV^\top - X))V + \lambda UV^\top V = 0 \, .$$

Again, rearranging and employing proposition 3.3 gives

$$\begin{aligned}
\text{vec}&(X^\top(W \odot X)V) \\
&= \text{vec}(X^\top(W \odot (XUV^\top))V + UV^\top V) \\
&= (H_D(U, 0) + \lambda(V^\top V \otimes I))\,\text{vec}(U) \\
&= H_D(U, \lambda)\,\text{vec}(U) \, .
\end{aligned}$$

Thus when employing dropout style regularization as in eq. (3), we have that

$$U_{\text{dropout}} = H_D(U, \lambda)^{-1}\,\text{vec}(X^\top(W \odot X)V) \, . \quad (12)$$

Now taking the derivative of eq. (2) with respect to $U$ and repeating the above calculations gives us the following closed-form solution for $U$ when employing a weight decay style of regularization as in eq. (2)

$$U_{\text{weight decay}} = H_W(U, \lambda)^{-1}\,\text{vec}(X^\top(W \odot X)V) \, . \quad (13)$$

Repeating the above calculations but for the matrix $V$ gives

$$V_{\text{dropout}} = H_D(V, \lambda)^{-1}\,\text{vec}((W^\top \odot X^\top)XU) \quad (14)$$

for eq. (3) and

$$V_{\text{weight decay}} = H_D(V, \lambda)^{-1}\,\text{vec}((W^\top \odot X^\top)XU) \quad (15)$$

for eq. (2). Thus we have derived the closed form solutions for the minimization problems defined in eq. (4), eq. (3) and eq. (2).

With the closed form solutions for eqs. (2) to (4) in place, we now turn our attention to computation

## 3.3 Computational Considerations

In this section, we address some of the computational issues with the closed form solutions given in the previous sections. The main issue with our closed form solution is that the memory requirements for computing the gram matrices $H$, $H_D$ and $H_W$ is often prohibitively large. For example, suppose one wants to compute $H_D(U, \lambda)$ on the MovieLens 20 Million (ML- 20M) dataset (Harper and Konstan [11]), which has $136, 677$ users and $20, 108$ items (movies) with around 10 million interactions. A quick calculation gives us that the sparse "binarized" click matrix for ML-20M, i.e. $X$, has density $0.003$, meaning about $0.3$ percent of the matrix $X$ is nonzero. If we let $d = 10$ in eqs. (2) and (3) and let $V \in \mathbb{R}^{|\mathcal{I}| \times}$ be a dense matrix,

**Algorithm 1:** Preconditioned Conjugate Gradient Method

**Input:** $A$: symmetric positive definite matrix, $b$: vector, $x_0$:
initial guess, $M^{-1}$: preconditioner, $\epsilon$: tolerance,
$max\_iter$: maximum iterations

**Output:** $x$: solution to $Ax = b$

$r_0 \leftarrow b - Ax_0$;

$z_0 \leftarrow M^{-1}r_0$;

$p_0 \leftarrow z_0$;

**for** $k \leftarrow 0$ **to** $max\_iter - 1$ **do**

  **if** $\|r_k\|_2 \leq \epsilon$ **then**

    **return** $\underline{x_k}$;

  **end**

  $\alpha_k \leftarrow (r_k^T z_k)/(p_k^T A p_k)$;

  $x_{k+1} \leftarrow x_k + \alpha_k p_k$;

  $r_{k+1} \leftarrow r_k - \alpha_k A p_k$;

  $z_{k+1} \leftarrow M^{-1} r_{k+1}$;

  $\beta_k \leftarrow (r_{k+1}^T z_{k+1})/(r_k^T z_k)$;

  $p_{k+1} \leftarrow z_{k+1} + \beta_k p_k$;

**end**

**return** $\underline{x_k}$

then computing the matrix $V \otimes X$, which is needed to compute $H_U$, would require more than $1,000$GB of storage.

## 4 An Efficient Algorithm

In this section, we give an efficient algorithm for solving eqs. (11) to (15). The design is based on the following two key insights:

- The regularized Gram matrices ($H, H_D$ and $H_W$) corresponding to the aforementioned equations are symmetric positive definite.
- While computing and storing these matrices is prohibitively expensive, computing and storing their product with a vector is (relatively) inexpensive, and indeed is tractable for problems of the scale considered in our experiments.

In order to understand the second insight more concretely, define a matrix $P \in \mathbb{R}^{|\mathcal{I}| \times |\mathcal{I}|}$ and let $p = \text{vec}(P)$. Then eq. (5) of proposition 3.3, reveals the equivalence between computing $H(B, \lambda)p$ and

$$\text{vec}\left(X^\top (W \odot (XP)) + \lambda P\right).$$

If instead we have $P \in \mathbb{R}^{\mathcal{I} \times d}$ then computing $H_D(U, \lambda)p, H_D(V, \lambda)p$, $H_W(U, \lambda)p$ and $H_W(V, \lambda)p$ can be done efficiently by invoking eqs. (5) and (6) of proposition 3.3 in an analogous way.

Therefore, since the Gram matrices are symmetric positive definite and their products with arbitrary vectors are inexpensive to compute, the natural algorithm for solving eqs. (11) to (15) is the conjugate gradient method of Hestenes and Stiefel [12] with preconditioning (Axelsson [1]), which is detailed in algorithm 1. The conjugate gradient method is an iterative method for solving positive definite linear systems of equations. The main computational bottleneck in algorithm 1 is computing matrix vector products but, as we argued above, this can be done (relatively) cheaply. Another pleasing property of algorithm 1 is that it converges to the solution of eqs. (11) to (15) in *at most* $|\mathcal{I}|d$ iterations, see Theorem 5.1 of Nocedal and Wright [20].

However, algorithm 1 may identify a solution in fewer than $|\mathcal{I}|d$ iterations when the regularized Gram matrices ($A$ in the algorithm) are well behaved. Specifically, let the conditioning number of $A$ be defined by $\kappa(A) = \lambda_{\max}(A)/\lambda_{\min}(A)$, where $\lambda_{\max}(A)$ and $\lambda_{\min}(A)$ are the maximum and minimum eigenvalues of $A$ respectively. Then the values of $x_k$ for $k = 0, \dots, max\_iter - 1$, as computed by algorithm 1, satisfy

$$\|x_k - x\|_A \leq 2 \exp\left(-\frac{2k}{\kappa(A)}\right) \|x_0 - x\|_A.$$

Thus algorithm 1 converges exponentially fast to the solution of any positive definite linear system with a reasonably small conditioning number.

In order to control the conditioning number of the various Gram matrices, we will employ a preconditioner $M^{-1}$ which will help to speed up the convergence of algorithm 1 without increasing the computational complexity. At a high level, preconditioning transforms the linear system $Ax = b$ to the linear system $M^{-1}Ax = M^{-1}b$ for an invertible matrix $M$. The goal is to choose $M$ so that $M^{-1}A$ has a lower conditioning number than $A$, and thus the conjugate gradient method has a fast convergence rate on the transformed system. Empirically we found the Gram matrix corresponding to $W = 1$, e.g. for $H(B, \lambda)$ we would use $M^{-1} = ((I_{|\mathcal{I}|} \otimes X^\top X) + \lambda I_{|\mathcal{I}|})^{-1} = (I_{|\mathcal{I}|} \otimes (X^\top X + \lambda I_{|\mathcal{I}|})^{-1})$. This is because the conditioning number of $M^{-1}H$ is of order $\max(W)$, which in our experiments is controlled by $\alpha$. Also, the $M^{-1}$ corresponding to eqs. (11) to (15) can be computed efficiently via proposition 3.3.

## 5 Numerical Experiments

In this section, we apply the models derived in Section 3, as well as the celebrated weighted matrix factorization model (Hu et al. [14], Pan et al. [21]), to collaborative filtering and recommendation problems. We focus on weighted linear autoencoders to better understand the interplay between *weighting* ($W$), *regularization* ($\lambda$) and *model-dimension* ($d$).

### 5.1 Experimental Setup

We follow the experimental setup detailed in Liang et al. [18], using their publicly available code, as well as the same three standard datasets:

- MovieLens 20 Million (ML-20M) (Harper and Konstan [11]) which consists of $136,677$ users, $20,108$ items and about 10 million interactions,
- Netflix Prize (Netflix) (Bennett and Lanning [3]) which consists of $463,435$ users, $17,769$ items and about 57 million interactions,
- Million Song Data (MSD) (Bertin-Mahieux et al. [4]) which consists of $571,355$ users, $41,140$ items and about 34 million interactions.

We also employ the same data preprocessing as in Liang et al. [18]. The user-by-item matrix $X$ is "binarized" with 1 denoting that a user interacted with an item (e.g., the user watched a movie and gave a rating above 3) and 0 denoting that a user has not interacted with an item. In this work we compare the following five models:

(1) Asymmetrical Weighted Matrix Factorization (AWMF) with weight decay (Steck [31]). This corresponds to the optimization

**Table 1: Ranking accuracies (Recall@20, Recall@50 and nDCG@100 as given in Liang et al. [18]) for linear autoencoders of model-rank $d = 1000$ trained with different regularizers and weightings $\alpha$, on three standard datasets, ML-20M, Netflix, and MSD, each with standard errors of $0.002, 0.001, 0.001$ respectively. Note that we report that results for both the best performing weighted model (and it's weight $\alpha$) and best performing unweighted model $\alpha = 1$. Note that each model has an additional hyperparameter $\lambda$ which controls the various $\ell_2$ regularizers.**

| Model | ML-20M | | | | Netflix | | | | MSD | | | |
|---|---|---|---|---|---|---|---|---|---|---|---|---|
| | Recall @20 | Recall @50 | nDCG @100 | $\alpha$ | Recall @20 | Recall @50 | nDCG @100 | $\alpha$ | Recall @20 | Recall @50 | nDCG @100 | $\alpha$ |
| 1. $\left\|\sqrt{W}\odot(X-XUV^\top)\right\|_F^2 + \lambda(\|U\|_F^2+\|V\|_F^2)$ | 0.344 | 0.476 | 0.378 | 2 | 0.311 | 0.395 | 0.345 | 5 | 0.228 | 0.314 | 0.277 | 10 |
| $\left\|X-XUV^\top\right\|_F^2 + \lambda(\|U\|_F^2+\|V\|_F^2)$ | 0.348 | 0.466 | 0.381 | 1 | 0.302 | 0.376 | 0.334 | 1 | 0.189 | 0.251 | 0.230 | 1 |
| 2. $\left\|\sqrt{W}\odot(X-UV^\top)\right\|_F^2 + \lambda(\|U\|_F^2+\|V\|_F^2)$ | 0.377 | 0.512 | 0.409 | 3 | 0.292 | 0.376 | 0.329 | 2 | 0.268 | 0.371 | 0.322 | 21 |
| $\left\|X-UV^\top\right\|_F^2 + \lambda(\|U\|_F^2+\|V\|_F^2)$ | 0.328 | 0.443 | 0.360 | 1 | 0.318 | 0.406 | 0.352 | 1 | 0.189 | 0.251 | 0.231 | 1 |
| 3. $\left\|\sqrt{W}\odot(X-XUV^\top)\right\|_F^2 + \lambda\left\|UV^\top\right\|_F^2$ | 0.355 | 0.491 | 0.388 | 2 | 0.337 | 0.419 | 0.370 | 2 | 0.227 | 0.311 | 0.277 | 10 |
| $\left\|X-XUV^\top\right\|_F^2 + \lambda\left\|UV^\top\right\|_F^2$ | 0.378 | 0.511 | 0.407 | 1 | 0.337 | 0.417 | 0.369 | 1 | 0.216 | 0.296 | 0.266 | 1 |
| 4. $\left\|\sqrt{W}\odot(X-XUV^\top)\right\|_F^2 + \lambda(\|XU\|_F^2+\|V\|_F^2)$ | 0.356 | 0.491 | 0.383 | 2 | 0.263 | 0.355 | 0.303 | 2 | 0.231 | 0.322 | 0.283 | 10 |
| $\left\|X-XUV^\top\right\|_F^2 + \lambda(\|XU\|_F^2+\|V\|_F^2)$ | 0.326 | 0.441 | 0.358 | 1 | 0.314 | 0.401 | 0.348 | 1 | 0.190 | 0.251 | 0.231 | 1 |
| 5. Full Rank $\left\|\sqrt{W}\odot(X-XB)\right\|_F^2 + \lambda(\|B\|_F^2)$ | 0.360 | 0.499 | 0.389 | 2 | 0.338 | 0.421 | 0.370 | 2 | 0.277 | 0.377 | 0.338 | 2 |
| $\left\|X-XB^\top\right\|_F^2 + \lambda(\|B\|_F^2)$ | 0.376 | 0.511 | 0.407 | 1 | 0.335 | 0.417 | 0.368 | 1 | 0.284 | 0.384 | 0.344 | 1 |

**Table 2: Ranking accuracies for linear autoencoders of model-rank $d = 100$.**

| Model | ML-20M | | | | Netflix | | | | MSD | | | |
|---|---|---|---|---|---|---|---|---|---|---|---|---|
| | Recall @20 | Recall @50 | nDCG @100 | $\alpha$ | Recall @20 | Recall @50 | nDCG @100 | $\alpha$ | Recall @20 | Recall @50 | nDCG @100 | $\alpha$ |
| 1. $\left\|\sqrt{W}\odot(X-XUV^\top)\right\|_F^2 + \lambda(\|U\|_F^2+\|V\|_F^2)$ | 0.339 | 0.472 | 0.374 | 2 | 0.309 | 0.392 | 0.343 | 5 | 0.167 | 0.241 | 0.210 | 10 |
| $\left\|X-XUV^\top\right\|_F^2 + \lambda(\|U\|_F^2+\|V\|_F^2)$ | 0.348 | 0.466 | 0.381 | 1 | 0.302 | 0.376 | 0.334 | 1 | 0.129 | 0.182 | 0.163 | 1 |
| 2. $\left\|\sqrt{W}\odot(X-UV^\top)\right\|_F^2 + \lambda(\|U\|_F^2+\|V\|_F^2)$ | 0.364 | 0.497 | 0.399 | 6 | 0.316 | 0.396 | 0.348 | 3 | 0.178 | 0.265 | 0.223 | 21 |
| $\left\|X-UV^\top\right\|_F^2 + \lambda(\|U\|_F^2+\|V\|_F^2)$ | 0.319 | 0.434 | 0.350 | 1 | 0.312 | 0.388 | 0.344 | 1 | 0.126 | 0.179 | 0.160 | 1 |
| 3. $\left\|\sqrt{W}\odot(X-XUV^\top)\right\|_F^2 + \lambda\left\|UV^\top\right\|_F^2$ | 0.320 | 0.454 | 0.361 | 2 | 0.316 | 0.396 | 0.348 | 2 | 0.167 | 0.239 | 0.208 | 10 |
| $\left\|X-XUV^\top\right\|_F^2 + \lambda\left\|UV^\top\right\|_F^2$ | 0.336 | 0.466 | 0.368 | 1 | 0.318 | 0.395 | 0.348 | 1 | 0.132 | 0.186 | 0.167 | 1 |
| 4. $\left\|\sqrt{W}\odot(X-XUV^\top)\right\|_F^2 + \lambda(\|XU\|_F^2+\|V\|_F^2)$ | 0.344 | 0.475 | 0.374 | 2 | 0.304 | 0.384 | 0.338 | 2 | 0.171 | 0.247 | 0.213 | 20 |
| $\left\|X-XUV^\top\right\|_F^2 + \lambda(\|XU\|_F^2+\|V\|_F^2)$ | 0.315 | 0.430 | 0.346 | 1 | 0.312 | 0.388 | 0.343 | 1 | 0.128 | 0.180 | 0.161 | 1 |

problem detailed in eq. (2). The regularization here encourages the entries of $U$ and $V$ to be close to zero.

(2) Weighted Matrix Factorization (WMF) (Hu et al. [14], Pan et al. [21]). A linear model that learns a user embedding matrix $U$ and item embedding matrix $V$ that minimize $\min_{U,V} \|\sqrt{W}\odot$ $(X-UV^\top)\| + \lambda(\|U\|_F^2+\|V\|_F^2)$. This method also employs a weight-decay style of regularization, encourages the entries of $U$ and $V$ to be near zero.

(3) Asymmetrical Weighted Matrix Factorization (AWMF) with dropout (Cavazza et al. [6], Steck [31]). This corresponds to the

Table 3: Ranking accuracies for linear autoencoders of model-rank $d = 10$.

| | Model | ML-20M | | | | Netflix | | | | MSD | | | |
|---|---|---|---|---|---|---|---|---|---|---|---|---|---|
| | | Recall @20 | Recall @50 | nDCG @100 | $\alpha$ | Recall @20 | Recall @50 | nDCG @100 | $\alpha$ | Recall @20 | Recall @50 | nDCG @100 | $\alpha$ |
| 1. | $\left\|\sqrt{W}\odot(X-XUV^\top)\right\|_F^2 + \lambda(\|U\|_F^2+\|V\|_F^2)$ | 0.305 | 0.426 | 0.338 | 5 | 0.259 | 0.339 | 0.293 | 2 | 0.086 | 0.134 | 0.114 | 20 |
| | $\left\|X-XUV^\top\right\|_F^2 + \lambda(\|U\|_F^2+\|V\|_F^2)$ | 0.297 | 0.415 | 0.332 | 1 | 0.258 | 0.335 | 0.290 | 1 | 0.074 | 0.111 | 0.099 | 1 |
| 2. | $\left\|\sqrt{W}\odot(X-UV^\top)\right\|_F^2 + \lambda(\|U\|_F^2+\|V\|_F^2)$ | 0.305 | 0.427 | 0.339 | 3 | 0.258 | 0.340 | 0.293 | 3 | 0.082 | 0.130 | 0.111 | 21 |
| | $\left\|X-UV^\top\right\|_F^2 + \lambda(\|U\|_F^2+\|V\|_F^2)$ | 0.289 | 0.401 | 0.322 | 1 | 0.259 | 0.337 | 0.292 | 1 | 0.074 | 0.110 | 0.099 | 1 |
| 3. | $\left\|\sqrt{W}\odot(X-XUV^\top)\right\|_F^2 + \lambda\left\|UV^\top\right\|_F^2$ | 0.303 | 0.425 | 0.336 | 5 | 0.260 | 0.340 | 0.293 | 2 | 0.085 | 0.129 | 0.111 | 10 |
| | $\left\|X-XUV^\top\right\|_F^2 + \lambda\left\|UV^\top\right\|_F^2$ | 0.298 | 0.415 | 0.331 | 1 | 0.258 | 0.334 | 0.290 | 1 | 0.075 | 0.110 | 0.098 | 1 |
| 4. | $\left\|\sqrt{W}\odot(X-XUV^\top)\right\|_F^2 + \lambda(\|XU\|_F^2+\|V\|_F^2)$ | 0.293 | 0.412 | 0.324 | 2 | 0.260 | 0.339 | 0.294 | 2 | 0.083 | 0.129 | 0.110 | 20 |
| | $\left\|X-XUV^\top\right\|_F^2 + \lambda(\|XU\|_F^2+\|V\|_F^2)$ | 0.290 | 0.407 | 0.321 | 1 | 0.260 | 0.338 | 0.293 | 1 | 0.074 | 0.110 | 0.098 | 1 |

optimization problem detailed in eq. (3). The regularization here encourages $U$ to be orthogonal to $V^\top$, that is $UV^\top \approx 0$. This style of weighting was inspired by the regularizer originally employ by EASE in Steck [30].

(4) Asymmetrical Weighted Matrix Factorization (AWMF) with "data/weight decay". This model employs the regularization used in WMF with the training objective used by AWMF, i.e., $\min_{U,V} \|\sqrt{W}\odot(X-XUV^\top)\| + \lambda(\|XU\|_F^2+\|V\|_F^2)$, where we let $XU$ be the user embedding in WMF. This encourages learning a matrix $U$ such that the elements of $XU$ are close to zero.

(5) Full Rank AWMF. This corresponds to the optimization problem detailed in eq. (4). This model learns a single full rank matrix $B$ and can be seen as the model induced by let $U$ and $V$ in AWMF be full rank.

Each numbered model in the above list corresponds to the model in tables 1 to 3. We determine the optimal training hyper-parameter $\lambda$ and $\alpha$ by performing a grid search. All models reported in tables 1 to 3 swept over $\lambda \in \{0.0001, 0.01, 1, 100, 10,000\}$ and $\alpha \in \{1, 2, 5, 10, 20\}$ similar to the sweep used by Liang et al. [18] when computing the optimal hyperparameters for WMF. If the highest validation performance occurred either at $\lambda = \{0.0001, 10,000\}$ or at $\alpha = 20$ then we increased the grid until validation performance dropped. All models in tables 1 to 3 were implemented using Python, Numpy and Scipy. Models 1, 3, 4, 5 in table 1 were implementing using algorithm 1 and preconditioner corresponding to their respective unweighted Gram matrix $G$. WMF (model 2) was computed using the publicly available code provided by the authors[4] of Liang et al. [18] though we remark WMF could also be implemented with algorithm 1. To evaluate the ranking accuracy of the various learned models on the withheld test set, we employed the evaluation scheme given in Liang et al. [18] as well as the normalized

[4]Their code adds one to all the values of $\alpha$ as is consistent with Hu et al. [14]. This is reflected in tables 1 to 3

Discounted Cumulative Gain (nDCG@100) and Recall (@20 and @50) detailed therein.

## 5.2 Experimental Results

Tables 1 to 3 gives the ranking accuracies obtained by various models with matrix ranks $d = 1000, 100$ and $10$, respectively, and for the various regularization and weighting schemes detailed in our paper. Table 1 also contains the ranking accuracies for the full rank model where $d = |\mathcal{I}|$.

First, let us consider the main finding of our paper: that (one of) the best performing ranking accuracies on each of the three datasets all belong to *unweighted methods*. To clarify the previous statement, unweighted methods either get the best performance or have performance within the standard error of the best performing weighted method. We can see from table 3 that weighting generally benefits models with lower rank. As the model-rank gets larger, weighting begins to hurt the performance of model in terms of the ranking accuracies as shown in table 1. For the full rank model, we see that the unweighted model either beats or matches the performance of its weighted variant across all three datasets and ranking metrics. We also note that $\alpha = 2$ (the smallest weights we grid-searched) performed best for the weighted full rank model. That the ranking accuracy increases with model rank is consistent with the findings in Steck [31], who highlight a similar trend for unweighted models.

The results on MSD might appear to be an outlier at first glance as all the weighted low-rank models outperform the unweighted ones. However, the performance of full-rank model provides a fuller picture: i) The unweighted full-rank model outperforms the best low-rank model (WMF with $d = 1000$) by a wide margin; ii) The significantly more rapid performance degradation with smaller $d$ suggests the underlying structure of the MSD dataset is much more complex than the other two. This is in line with the observation in Steck and Garcia [32], Steck and Liang [33] that MSD has a

much longer-tailed distribution. This is also consistent with our observation that weighting matters less when the capacity of a low-rank model is getting close to the full-rank one.

Now let us turn our attention to the different $\ell_2$ regularizers. Our findings corroborate those of Steck [31], Steck and Garcia [32]. Namely that dropout (line 3 table 1) seems to outperform the weight decay (line 1 table 1) in terms of ranking accuracy for the unweighted models for larger model-ranks. When the model-rank is smaller as in tables 2 and 3 our findings seems to suggest that neither method is conclusively better. Specifically, when the model-rank is $d = 100$, dropout outperforms weight decay on Netflix but under performs on ML-20M.

The hybrid regularizer given in line 4 of the tables was chosen to help understand why weighting benefits WMF. As mentioned earlier, if we let $XU$ be the user embedding matrix in WMF then we end up with the hybrid regularizer. This regularizer encourages learning a $U$ such that $XU \approx 0$. Thus we expect up weighting the nonnegative elements of $X$ to benefit this method as we do not want to predict zero for the nonnegative entries of $X$. For this method, the best choices of $\lambda$ where values very close to zero, i.e., $10^{-6}$ and $10^{-8}$. This is expected as we do not want to predict zero when $X_{ui} = 1$. As $\lambda$ took larger values, it was often very beneficial to the method to use larger weights to compensate for the larger values of $\lambda$. We believe these experimental findings corroborate our intuition that weighting benefited WMF precisely because WMF was implicitly regularizing the user embedding to predict zero when $X$ took nonnegative values.

In summary, we find that as the model-rank of AWMF grows the effects of up weighting the positive values of $X$ is detrimental to the performance of the model as measured by ranking-accuracy. Weighted does indeed help WMF, which is in accordance with the previous literature (Hu et al. [14], Pan et al. [21], Saito et al. [26], Steck [27]). We believe our experimental findings for AWMF with data/weight decay give some intuition into why this phenomena seems to occur.

## 6 Related Work

Weighted matrix factorization for implicit feedback data was introduced in Hu et al. [14], Pan et al. [21].Based on this seminal work, in many practical applications, the observed/positive user-item interactions were up-weighted relative to the unobserved/missing user-item interactions, all with the same weight, hence resulting in a single scalar weight as an additional hyperparameter that can be tuned during training by optimizing the (unweighted) ranking metric on the validation set. Over the years, it has become common wisdom that up-weighting the positive user-item interactions is key to obtaining improved recommendation accuracy. It also made intuitive sense, as up-weighting the positive user-item interactions can be understood in several ways: (i) it corresponds to a reduced uncertainty of the positive data points, and (ii) due to the sparsity of the positive data points, up-weighting them makes the data more balanced, a common practice when dealing with unbalanced data sets. Instead of up-weighting the positive samples, the negative data-points are down-sampled for computational efficiency in practice. A recent paper proposes a combination of sampling and weighting (Petrov and Macdonald [23]).

The MF model was typically trained iteratively, either by stochastic gradient descent or alternating least squares (e.g., Hu et al. [14], Pan et al. [21], Rendle et al. [25]). As we noticed in our current work, finding a near-optimum solution is not trivial even for such bi-linear models, but it is crucial for better understanding the effect of weighting the data-points, as shown in this paper.

While the matrix factorization model decomposes the user-item interaction matrix into a product of low-rank user and item latent factors, similar to (unweighted) singular value decomposition (or pureSVD in the recommendation literature (Cremonesi et al. [7]), other variants were proposed in the literature as well, most notably asymmetric MF (Paterek [22]), where two (different) item-factor matrices are learned, reducing the number of model-parameters considerably (in case of $|\mathcal{U}| \gg |\mathcal{I}|$). Training this model with a weighting scheme was also typically beneficial.

In contrast, the full-rank models called SLIM (Ning and Karypis [19]), and its simplified variant called EASE (Steck [30]) are trained on unweighted data, yet obtain competitive results. These models also use a different variant of L2-norm regularization, related to dropout/denoising instead of weight decay (Steck [31]).

Finally, it is argued in Byrd and Lipton [5] that the effect of weighting diminishes as the capacity of the (deep learning) model increases such that is able to fit the data, e.g, in a classification task the data may become separable. Interestingly, this paper also argues that weight-decay regularization may prevent large-norm solutions, which hence may require weighted training–in contrast, dropout-regularization does not have this effect and hence may require less/no weighting.

All this related work indicates that there is more to uncover as to how the weighting is affected by the model capacity as well as the regularization, which motivated this paper.

## 7 Conclusion

Our systematic study of weighting schemes for matrix factorization models on implicit feedback data revealed several key findings.

Unweighted methods often outperform their weighted counterparts, especially as model capacity increases. The benefits of weighting diminish with increasing model rank. The choice of regularization scheme interacts with weighting, with dropout-style regularization generally outperforming weight decay for unweighted higher-rank models as reported by Steck [31].

We provided efficient algorithms for optimizing weighted objectives for various matrix factorization formulations. These results challenge the conventional wisdom of upweighting observed interactions, particularly for higher-capacity models. Our findings have important implications for recommender system design, suggesting that practitioners should carefully evaluate the necessity of weighting for their specific models and datasets.

Future work could explore the theoretical foundations of these observations and investigate their applicability to deep learning models for recommendation. Overall, our work underscores the need to reassess common assumptions in recommender systems as model architectures evolve.

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
