# OpenReview forum: "Does weighting improve matrix factorization for recommender systems?"
_ACM.org/TheWebConf/2025/Conference — WWW 2025 Poster_

### Official Review · Reviewer_kCin · 2024-11-11

**Novelty:** 4
**Technical Quality:** 6

**Review:**

This paper explores recommender systems that rely on implicit feedback—interactions like clicks, views, and purchases—rather than explicit feedback, such as ratings, to predict items that users might enjoy. A primary challenge for matrix factorization with implicit feedback is managing the zeros in the user-item interaction matrix.

To address this ambiguity, common practice involves weighting the data to emphasize observed interactions, under the assumption that these interactions carry more reliable information about user preferences. However, the authors of this paper challenge this convention by showing, via experimental evidence, that unweighted models often perform better than their weighted counterparts on standard benchmarks, particularly in ranking accuracy.

The authors also introduce efficient algorithms for minimizing weighted objectives across various matrix factorization models. Finally, they conduct comprehensive experiments to analyze the interaction between weighting, regularization, and model capacity. Experimental results show that weighting is sometimes advantageous for simpler models but is not universally beneficial.

This work fits very well with the topics of interest of the “User modeling, personalization, and recommendation” track at The Web Conference 2025, especially the “ML for personalized search and recommendations” category.

This paper addresses an interesting problem, and the authors propose a sound evaluation of weighting strategies of matrix factorization model for implicit feedback, providing technically valuable insights. I appreciated the fact this paper provided both empirical and theoretical analysis, backed by extensive experimentation, which provides insights the role of weighting schemes across various model capacities. Additionally, the paper introduces efficient algorithms for minimizing weighted objectives.

I will now provide aspects that, in my opinion, would need to be revised, along with suggestions on how to possibly deal with them. When talking about future revisions of this manuscript, it might be for the camera-ready version of the paper or for an evolution of this study, besides this submitted manuscript. With my comments, I hope to provide constructive criticism to the authors.

Regarding the type of contribution, conceptually, this looks more like a reproducibility paper rather than a research one. I understand there are also technical contributions, given by the optimization techniques, but I believe a full research paper, in a conference at the caliber of WWW, requires stronger technical contributions. I would recommend the authors to consider targeting reproducibility tracks, which are available at similar conferences, such as SIGIR, RecSys, or ECIR. In the alternative, the Perspective track at SIGIR, where classic paradigms are challenged, might also be a possible outlet for this study.

Also concerning the contributions, I’m afraid the impact of this study might be limited. Indeed, recent advances in recommender systems go beyond Matrix Factorization. Again, I believe a reproducibility track would probably be better welcoming studies of this type. In the alternative, strengthening the presentation of the impact of the contributions would help.

Moving to the reproducibility, I’m afraid it might be challenged by the absence of the source code. Indeed, the optimization function and the implementation details are quite complex. I recommend the authors to integrate the source code of their approach and of the evaluation framework when revising their manuscript.

In conclusion, this paper makes valuable contributions by questioning conventional weighting practices in matrix factorization for implicit feedback and demonstrating that unweighted approaches can sometimes outperform their weighted counterparts in recommendation accuracy. The authors’ empirical and theoretical analyses, along with their efficient algorithms for optimizing weighted objectives, provide technically sound insights. However, the study’s reliance on matrix factorization, an approach somewhat surpassed by more advanced techniques, may limit its impact, especially in the context of a research track. Including source code is also suggested, so as to enhance the paper’s reproducibility and practical utility. Strengthening the discussion of the study's relevance to the potential real-world impact could also help broaden its appeal. I hope these comments will be useful for the authors and their future advances in this interesting research area.

COMMENTS AFTER THE AUTHORS' RESPONSE. I thank the authors for addressing my points. I appreciate their willingness to release the source code once the paper is published. I still have doubts regarding whether the technical contributions are enough for a full paper at WWW or if they would fit a different type of track, but I will discuss with the other reviewers the pros and cons of this study.

**Questions:**

Q1. Will you be releasing the source code for your optimized algorithms and evaluation framework? If so, could you provide details on when and where it will be available?
Q2. Why do you think a study on the optimization of Matrix Factorization models makes sense in the current landscape, given the development of algorithms that are both more effective and more efficient?
Q3. Would you consider this study as better fitting a reproducibility or perspective study?

**Ethics Review Description:**

I don’t have any ethical concerns.

**Reviewer Confidence:**

4: The reviewer is certain that the evaluation is correct and very familiar with the relevant literature

**Scope:**

3: The work is somewhat relevant to the Web and to the track, and is of narrow interest to a sub-community

---

### Official Review · Reviewer_hGmx · 2024-11-29

**Novelty:** 4
**Technical Quality:** 5

**Review:**

Focusing on top-N recommendations with implicit feedback, the paper studies various matrix factorization algorithms and their weighted versions, and surprisingly finds that *unweighted methods outperform their weighted counterparts*. Also, the paper analyzes the interplay between *weighting ($W$), regularizers ($\lambda$) and model dimension ($d$)* , and proposes *a novel optimization algorithm* to tackle the challenges of finding efficient solutions.

Overall, this work has a well-structured experimental design, an in-depth analysis and a counterintuitive discovery. Moreover, the paper demonstrates impressive mathematical skills and provides detailed derivations.

However, the paper still lacks adequate explanations for some experimental details, and fails to provide the necessary intuitive evidence to support the main finding, namely, that *unweighted methods often outperform weighted ones*. I reckon this phrasing of the conclusion is somewhat over-polished.

**Pros**

**1. Well-structured experimental design and in-depth analysis.**

The experimental design in the paper is well-structured, utilizing datasets of different scales (ML-20M, Netflix, MSD) and various matrix factorization algorithms. Furthermore, the authors consider multiple hyperparameters, including $W$, $\lambda$ and $d$, and provide an in-depth analysis of the interplay between these factors based on the experimental results.

**2. Counterintuitive discovery.**

The authors find that the best-performing recommendation models are trained using unweighted methods instead of weighted ones. This discovery challenges the conventional wisdom, suggesting that practitioners should carefully evaluate the necessity of weighting, which calls for a reassessment of common assumptions in recommender systems.

**3. A novel efficient optimization algorithm with detailed derivations.**

The paper derives a closed-form solution using the properties of the *Kronecker* product and the *vectorization lemma*, and combines this with the *conjugate gradient method* to propose an efficient algorithm, filling the gap in optimization methods for various objectives.

Moreover, the paper provides a detailed exposition of the derivations. Specifically, in Section 3.1, it first presents the derivation for $\lambda = 0$, and then continues with the case for $\lambda > 0$ in Section 3.2, making it easier for readers to follow.

**Cons**

**1. Lacks adequate explanations for some experimental details.**

First, the paper does not provide the exact values of $\lambda$. In Section 5.2 (Page 8, Line 834), the authors analyze the value of $\lambda$, but there is an absence of $\lambda$ values in Tables 1,2,3. Although Section 5.1 (Page 7, Line 737) includes an illustration of the "grid" for $\lambda$, the lack of exact values still makes the argument unconvincing.

Second, in Section 5.2 (Page 7, Line 746), the authors mention that, in the experiment, *"WMF (Model 2) was computed using the publicly available code provided by prior work"*. However, since they also remark that *"WMF could also be implemented with Algorithm 1"*, I believe Algorithm 1 should be applied to WMF as well, to ensure experimental consistency. Otherwise, it should be clarified that the results from the two algorithms do not differ significantly.

**2. Fails to provide intuitive evidence supporting the main finding.**

According to the paper, the main finding is that *“one of the best performing ranking accuracies on each of the three datasets belongs to unweighted methods”* (Section 5.2, Page 7, Line 788). Indeed, by selectively analyzing the data in the tables, we can draw this conclusion. However, at first glance, looking at the entire table, most weighted methods actually outperform the unweighted ones, which makes it difficult to intuitively accept the conclusion.

In fact, the authors could add a small table that separately lists the best-performing weighted and unweighted methods for each of the three datasets, which would provide a clear visual comparison for the reader. Additionally, authors could highlight the results in the larger tables using bold text or "$\uparrow$" to make the comparison more prominent.

Another approach would be to conduct experiments with higher $d$, which could make the conclusion more evident.

In summary, I believe the evidence for the main finding in the paper is not presented in a visually intuitive manner, as it lacks additional experiments to support it.

**3. Over-polished**

The paper concludes that *"unweighted methods often outperform their weighted counterparts, especially as model capacity increases"* (Section 7, Page 8 Line 906). However, I think this conclusion is not intuitively obvious and may be somewhat over-polished (especially the use of the word *'often'*) .

As mentioned earlier, the paper lacks intuitive evidence for the main finding. Furthermore, by analyzing the data in Tables 1,2,3, more than 70% of the results show that weighted methods still outperform their unweighted counterparts. In fact, we can easily observe that as $d$ increases, the model performance improves. Although weighting is harmful now, the improvement in performance due to the higher $d$ remains significant. This actually emphasizes the critical role of $d$ in model performance, rather than arguing "to weight or not to weight".

Many of the descriptions of the main finding in the paper creates a sense of a groundbreaking discovery, which certainly makes the article more attention-grabbing. However, I believe this phrasing is not sufficiently supported by the evidence and is somewhat overhyped. A more cautious phrasing, such as *"weighting implicit data is not always optimal"*, might be more appropriate.

**4. Inadequate discussion on related work**

In line 221 and line 229, the authors claim that "the challenge lies in efficient optimization with large-scale recommendation datasets." and "these solutions will be paired with the conjugate gradient method to produce efficient algorithms for the various optimization problems." which is quite confusing. There are many papers regarding large-scale matrix factorization for recommendation. Many of them are conducted based on gradient descent or Alternating Least Square over clusters or GPUs. Why the efficient solution is a challenge? I suggest the authors enrich the related work on large-scale matrix factorization for recommendation to make it more clear.

Based on the comments above, here are some suggestions for improvement:

- Add the specific values of $\lambda$ in Tables 1,2,3. (See Cons 1)
- Use Algorithm 1 to compute WMF in the experiments for consistency. (See Cons 1)
- Create a small table listing the best-performing weighted and unweighted methods on the three datasets to clearly present your main finding. (See Cons 2)
- Highlight the best performances in Tables 1,2,3 using bold text or markers like $\uparrow$ to visually guide the reader. (See Cons 2)
- Revise some overly embellished statements about the main conclusion or provide additional experimental data for better support. (See Cons 3)
- Enrich the related work on large-scale matrix factorization for recommendation to clearly explain why efficient matrix factorization is still a challenge.
- Erratum: In Section 2, is "$W_{ui} = (\alpha - 1)X_{ui} + 1$ for $\alpha > 0$" a typo? To increase the weight for positive feedback ($X_{ui} = 1$) , $\alpha > 1$ is required to ensure $W_{ui} (X_{ui} = 1) > W_{ui} (X_{ui} = 0)$. (See Page 2, Line 153)

**Questions:**

Questions are listed in cons.

**Reviewer Confidence:**

3: The reviewer is confident but not certain that the evaluation is correct

**Scope:**

4: The work is relevant to the Web and to the track, and is of broad interest to the community

---

### Official Review · Reviewer_Tayd · 2024-12-02

**Novelty:** 3
**Technical Quality:** 4

**Review:**

The authors conducted systematic experiments using various matrix factorization models and weighting schemes on three standard datasets: ML-20M, Netflix, and MSD. Their findings reveal that unweighted methods often outperform weighted counterparts, particularly as model capacity (rank) increases. They also show that the benefits of weighting diminish with higher model rank and that different regularization schemes, such as dropout and weight decay, interact differently with weighting. Additionally, the study introduces algorithms for optimizing weighted objectives, addressing areas that were previously unexplored. A key insight from this work is that the common practice of upweighting positive interactions can be detrimental for higher-capacity models, emphasizing the need for practitioners to carefully evaluate the necessity of weighting based on their specific model architecture and dataset. The study delivers a comprehensive empirical evaluation across diverse datasets and model variants, supported by a rigorous experimental methodology and clear ablation studies. It features well-structured mathematical derivations and proofs alongside a thorough comparison of weighting schemes and regularization methods. However, it could benefit from a deeper theoretical analysis of why weighting becomes less effective and lacks exploration of newer neural architectures. This work presents a novel challenge to established assumptions about weighting in matrix factorization and introduces efficient algorithms for weighted objectives. Its fresh perspective highlights the interaction between model capacity and weighting. Despite its innovation, the study remains rooted in existing matrix factorization frameworks, which limits its originality to some extent.

**Questions:**

1-While the empirical results are compelling, the paper lacks theoretical analysis explaining why weighting becomes less beneficial as model capacity increases. Could you provide mathematical intuition or theoretical guarantees for this phenomenon?

2-The study focuses exclusively on matrix factorization models. How do these findings extend to modern deep learning-based recommender systems? Would similar patterns emerge with neural architectures?

3-The proposed algorithms for weighted objectives rely on conjugate gradient methods. How does the computational cost compare to traditional alternating least squares approaches, especially for large-scale recommendations?

4-The paper mentions grid search for weighting parameter α and regularization λ. How sensitive are the conclusions to these hyperparameter choices? Could adaptive weighting strategies potentially bridge the gap between weighted and unweighted approaches?

5-The efficient algorithms provided require careful implementation of conjugate gradient methods with preconditioning. Can you provide more detailed guidance on implementing these methods in practice, particularly for handling numerical stability issues.

5-The study primarily focuses on uniform weighting schemes (α for positive interactions). Have you explored more sophisticated weighting strategies that might preserve the benefits of weighting while avoiding its drawbacks at higher model capacities?

**Reviewer Confidence:**

4: The reviewer is certain that the evaluation is correct and very familiar with the relevant literature

**Scope:**

4: The work is relevant to the Web and to the track, and is of broad interest to the community

---

### Official Review · Reviewer_vTbr · 2024-12-02

**Novelty:** 5
**Technical Quality:** 6

**Review:**

The paper investigates whether weighting is necessary in matrix factorisation-based recommender systems. There are two primary contributions: closed-form solutions for several matrix factorisation models and experiments to identify when weighting is beneficial and how it interactions with regularisation and model dimension in several real-world data sets.

Pros:

- Interesting research questions and approach.

- Combination of methodological and experimental contributions.

Cons:

- Results not as general as suggested in the text.

- Paper needs proof-reading in places.


I am the wrong person to comment on Sections 2, 3 and 4, so I will focus on the evaluation.

I found the presentation of results quite confusing: as we are looking at trends from d=10, 100, 1000, to full rank, it feels like it would be more natural to have a table per data set.

Section 5.2 has several statements about performance: "unweighted methods either get the best performance or have performance within the standard error of the best performing weighted method". Is this true? For example, the standard error for Netflix is only mentioned in Table 1's caption as 0.001, but the difference between weighted and unweighted full rank models are > 0.001 for recall@20, recall@50 and nDCG@100. Am I misinterpreting this statement?

And then it says "For the full rank model, we see that the unweighted model either beats or matches the performance of its weighted variant across all three datasets and ranking metrics." - this is not true for Netflix, only ML-20M and MSD.

In general, the results feel inconclusive:
  - unweighted only outperformed or tied weighted in model 3 (d=1000), model 3 (d=100) and the full rank model
  - unweighted only outperformed weighted in MSD with the full rank model
  - unweighted only outperformed weighted in 2/3 data sets with the full rank model (an additional data set would make it more conclusive)

This is a minor issue, but I would remove the more informal language, such as "the go-to choice", "there is more to be uncovered here", "the celebrated weighted matrix factorization model" and others.

**Questions:**

Please see questions in review related to my interpretation of Section 5.2 and the generality of the results.

**Reviewer Confidence:**

2: The reviewer is willing to defend the evaluation, but it is likely that the reviewer did not understand parts of the paper

**Scope:**

3: The work is somewhat relevant to the Web and to the track, and is of narrow interest to a sub-community

---

### Official Review · Reviewer_v7Lr · 2024-12-02

**Novelty:** 6
**Technical Quality:** 6

**Review:**

The paper investigates negative weighting schemes for matrix
factorization models (MF) for recommender systems. The authors
investigate three different factorization models, weighted MF,
asymmetrical MF and full-rank MF, with different regularization
penalties and with and without negative weighting. They develop
a preconditioned conjugate gradient solver for these models.
In experiments on three different well-known public datasets
they find that negative weighting in several cases harms the
performance instead of improving it.

The researched question, if negative weighting improves performance
in MF models, is clearly interesting. The derived conjugate gradient
solver is also an interesting contribution.

Major weaknesses:
- w1. The main claim of the paper, that
  "unweighted methods outperform their weighted counterparts"
  maybe needs to be formulated more carefully.
  - For example, in the highest rank experiment (tab. 1) the unweighted
  methods outperform their weighted counterpart in 2/9, 3/9, 3/9 (+3 ties),
  3/9 and 6/9 (+3 ties) cases, thus only for the last model, the full rank
  model, there is clear evidence for this statement.
  - Result tables maybe could be made easier to read by highlighting
  and counting wins.

**Questions:**

- q1. Do you have any hypothesis why weighting does help the
  weighted MF model, but not the full-rank model ?

**Reviewer Confidence:**

3: The reviewer is confident but not certain that the evaluation is correct

**Scope:**

4: The work is relevant to the Web and to the track, and is of broad interest to the community